# COVID-19 Clinical Features and Outcome in Italian Patients Treated with Biological Drugs Targeting Type 2 Inflammation

**DOI:** 10.3390/life14030378

**Published:** 2024-03-13

**Authors:** Giada Sambugaro, Elena Brambilla, Giulia Costanzo, Vera Bonato, Andrea Giovanni Ledda, Stefano Del Giacco, Riccardo Scarpa, Marcello Rattazzi, Elisabetta Favero, Francesco Cinetto, Davide Firinu

**Affiliations:** 1Department of Medical Sciences and Public Health, University of Cagliari and Azienda Ospedaliero Universitaria, SS 554-Bivio Sestu, 09042 Monserrato, Italy; 2Department of Medicine-DIMED, University of Padova, 35131 Padua, Italy; 3Rare Diseases Referral Center, Internal Medicine I, Ca’ Foncello Hospital, AULSS2 Marca Trevigiana, 31100 Treviso, Italy

**Keywords:** COVID-19, type 2 inflammation, biologicals, asthma

## Abstract

This is a multicentric investigation involving two Italian centers that examined the clinical course of COVID-19 in patients receiving biological therapy targeting type 2 inflammation and those not receiving biologicals. Since the beginning of the COVID-19 pandemic, the management of respiratory and allergic disorders and the potential impact of biological therapy in the most severe forms has been a point of uncertainty. Our multicentric investigation aimed to compare the clinical course of COVID-19 and the impact of vaccination in an Italian cohort of patients with atopic disorders caused by a type 2 inflammation, such as eosinophilic asthma, chronic rhinosinusitis with nasal polyposis (CRSwNP), atopic dermatitis (AD), and chronic spontaneous urticaria (CSU). A questionnaire was given to patients coming to our outpatient clinic for the first evaluation or follow-up visit, asking for the clinical characteristics of the infection, the ongoing therapy during the infection, any relevant change, and the patient’s vaccination status. We enrolled 132 atopic patients from two Italian centers; 62 patients were on biological therapy at the time of infection (omalizumab 31%, mepolizumab 26%, benralizumab 19%, and dupilumab 24%). The median age was 56 (IQR 22.8) for patients on biologicals and 48 (IQR 26.5) for those not on biologicals (*p* = 0.028). The two groups were comparable in terms of sex, body mass index (BMI), smoking history, and systemic oral corticosteroid use (OCS). There were no significant differences in non-biological therapy and comorbidity between the two groups. The patients not on biological therapy had a prevalence of 87% for asthma, 52% for CRSwNP, 10% for CSU, and 6% for AD. The patients on biologicals had a prevalence of 93% for asthma, 17% for CRSwNP, and 10% for CSU. In our work, we observed that mAbs targeting type 2 inflammation in patients with COVID-19 appeared to be safe, with no worsening of symptoms, prolongation of infection, or increase in hospitalizations. Between the two groups, there were no significant differences in the duration of swab positivity (*p* = 0.45) and duration of symptoms (*p* = 0.38). During COVID-19, patients on biologicals experienced a significant increase in common cold-like symptoms (*p* = 0.038), dyspnea (*p* = 0.016), and more, but not significant, asthma exacerbations, with no significant differences between the different biologicals. Regarding the vaccination status, we observed that there was an increased number of hospitalizations among unvaccinated patients in both groups, although the difference did not reach statistical significance. No patients on biologicals reported safety issues or adverse effects associated with the use of biological treatments during COVID-19. Our investigation showed that mAbs against type 2 inflammation given during Coronavirus Disease 2019 are safe and do not impact the clinical course or main outcomes. Therefore, we found no signals suggesting that anti-Th2 biological therapy should be discontinued during SARS-CoV-2 infection. Controlled studies and analysis, including data from registries and real-life studies, are required to draw firm conclusions regarding the safety or possible advantages that anti-type 2 mAbs could offer in particular clinical contexts, such as infections.

## 1. Introduction

The Coronavirus Disease 2019 (COVID-19) pandemic has negatively impacted the world since 2020; it is caused by acute respiratory syndrome coronavirus 2 (SARS-CoV-2) [1]. From the beginning of this pandemic, one of the most relevant concerns was the clinical management of chronic diseases, including patients with allergic and atopy-associated diseases, and how these pathologies and their treatments can impact the clinical course of COVID-19. The severity of manifestations in patients affected by COVID-19 mostly relies on the host immune response to the infection; the clinical features are diverse and range from asymptomatic to critical illness and death [2]. In the airway epithelium, the SARS-CoV-2 spike protein binds to angiotensin-converting enzyme 2 (ACE2) and cleaves the host transmembrane protease serine 2 protease (TMPRSS2). Other molecules involved in the invasion of SARS-CoV-2 are expressed at the epithelial level and immune cell level [3]. Atopy and type 2 inflammation are associated with reduced expressions of ACE2 in airway epithelial cells and, therefore, lower susceptibility to SARS-CoV-2 [1,4]. Caar et al. showed that type 2 cytokine expression is inversely correlated with ACE2 expression in nasal and airway epithelial cells from patients with asthma or allergic rhinitis, but positively correlated with TMPRSS2 expression [5]. Therefore, the introduction of biological therapy that can modulate type 2 inflammation has raised concerns about increased susceptibility to SARS-CoV-2 infection or its severity linked to the upregulation of ACE2.

At the beginning of the pandemic, concerns were raised that asthma patients could be at an increased risk of SARS-CoV-2 infection and disease severity, and subsequently, about the impact of different phenotypes of asthma on the risk of infection with SARS-CoV-2 and the severity of the disease. Data suggest that high Th2 inflammation may reduce the risk of SARS-CoV-2 infection and the severity of the disease, as opposed to increased risk in patients with low-Th2 asthma [6,7]. This has been explained in part by the decreasing gradient of angiotensin-converting enzyme-2 receptor from the upper to lower respiratory epithelium and by the fact that aeroallergen-sensitized asthmatics can have up to 50% reduction in ACE-2 receptor expression [8]. Adir et al. highlighted how the use of ICS in asthmatic patients during SAR-CoV-2 infection was safe, without worsening the severity of the disease or infectiousness. Similarly, the use of biological therapy for severe allergic and eosinophilic asthma does not increase the likelihood of contracting SARS-CoV-2 or worsen the symptoms of COVID-19. Conversely, repeated and chronic use of SCS before SARS-CoV-2 infection was correlated with adverse outcomes and lower survival rates [9]. However, limited data are available on this topic in other type 2 diseases, such as asthma, chronic rhinosinusitis with nasal polyposis (CRSwNP), atopic dermatitis (AD), and chronic spontaneous urticaria (CSU). In particular, there is uncertainty about the potential immunological interactions and treatment-emergent effects associated with these mAbs when administered during SARS-CoV-2 infection.

Thus, we aimed to describe the impact of biological therapy on the clinical course of COVID-19 among patients with type 2 diseases, analyzing the clinical characteristics of COVID-19, the duration of symptoms and swab positivity, and the incidence of exacerbation and hospitalization. We also evaluated how the vaccination impacted the outcomes.

## 2. Materials and Methods

### 2.1. Study Design

Our multicentric investigation aimed to compare the clinical course of COVID-19 and the impact of vaccination in an Italian cohort of patients with disorders caused by a type 2 inflammation, such as eosinophilic asthma, CRSwNP, AD, or CSU, in order to evaluate the impact of anti-Th2 biological therapy. We included patients who came to the hospital for a follow-up visit or first examination and who previously had experienced a SARS-CoV-2 infection, recording data using a questionnaire focused on COVID-19. 

### 2.2. Patient Recruitment

Adult subjects were recruited in two centers in Italy, Cagliari and Treviso. We consecutively included patients who came to our outpatient clinics for a follow-up visit or first examination and who previously had experienced SARS-CoV-2 infection. Each patient was asked to participate in data collection about the infection. The exclusion criteria for this study were (1) concurrent immune-mediated diseases beyond those listed in the inclusion criteria; (2) evidence of primary or secondary immunodeficiency; (3) relevant medication intake (e.g., immunosuppressants, continuous oral glucocorticoids > 10 mg/day). The enrolled patients had a type 2 inflammation phenotype, characterized by a history of atopy and/or high level of fractional exhaled nitric oxide (FeNO) and/or high levels of eosinophils. Participants with asthma met the Global Initiative for Asthma (GINA) criteria for mild, moderate, and severe asthma, including the presence of reversibility of airflow obstruction or airway responsiveness with a provocative concentration of methacholine resulting in a 20% decrease in Forced Expiratory Volume in the first second (FEV1) value. Participants with CRSwNP had a diagnosis made by facial CT scan and/or a Nasal Polyps Score over 7 points associated with an increased eosinophil count. Furthermore, some of these patients previously underwent surgery and required oral corticosteroids to control the disease. Patients with AD met the clinical major criteria of chronic recurring eczema, pruritus, typical locations and morphology of skin lesions, and personal and family history of atopy. We also included patients with CSU, considering the main findings of pathogenesis [10,11]. For data analysis, we stratified the patients into two groups: in one, the patients who, during COVID-19 infection, were on biological therapy against type 2 inflammation, and in the other, the control group, atopic patients not on biologicals and/or without a history of biological therapy use. The research was conducted in compliance with the Helsinki Declaration and was examined and approved by the “Comitato Etico per la Sperimentazione Clinica delle Province di Treviso e Belluno” (Protocol # 793/CE) and by AOU Cagliari Ethical Committee (SANI protocol, 23-07-2018 n. 15 and MANI protocol, 05-10-2022 n.5).

### 2.3. Biological Therapy

In this study, we included patients with allergy and respiratory disorders caused by a type 2 inflammation in which IgE or the interleukins IL-4, IL-5, and IL-13 play a key role in the pathogenesis. In recent decades, several mAbs have been approved in Europe for this immunological pathway, and we included patients who were receiving omalizumab, mepolizumab, benralizumab, and dupilumab. We did not include reslizumab because of its limited use in Italy. Omalizumab is an anti-IgE indicated for the treatment of severe persistent asthma caused by an allergy in patients from 6 years of age, CSU in patients aged 12 years or over not responding to antihistamines, and severe chronic rhinosinusitis with nasal polyps in adults [12]. Mepolizumab, an anti-IL-5, is approved for the treatment of severe eosinophilic asthma in patients aged 6 years and above, and recently, but not yet during our data collection period, also for CRSwNP in adult patients. Benralizumab, an anti-IL-5 receptor blocker, is available for the treatment of severe asthma in adults [13,14]. Dupilumab is indicated for the treatment of severe AD from 12 years of age, severe asthma in in patients aged 6 years and over, and CRSwNP in adults. It is also indicated for moderate-to-severe prurigo nodularis and eosinophilic esophagitis in adults and children above 12 years of age [15,16].

### 2.4. Questionnaire

A standardized questionnaire, shown in the Appendix A, was administered during outpatient visits (or integrated by phone for missing data), and data regarding COVID-19-related symptoms, hospital admission for the virus, and diagnostic testing through nasopharyngeal swab (PCR) were obtained. Patients’ data were collected between April 2022 and January 2023. Data about positivity for SARS-CoV-2 were collected from both symptomatic and asymptomatic people, in the latter case as part of screening programs such as planned health procedures or hospitalization. We also obtained data on bio-anthropometric characteristics, comorbidities, and therapies for the type 2 condition during the infection and their change, exacerbations of the baseline type 2 conditions, the vaccination status at the time of COVID-19 infection, and the clinical characteristics of the infection, such as duration of swab positivity and type of symptoms. The latter ranged from an asymptomatic phenotype to hospitalization or need of oxygen mask or ventilation.

### 2.5. Statistical Analysis

Categorical variables are presented as *n* (%), whereas numerical variables are presented as median (interquartile ranges [IQR]), as data were not normally distributed. The normality of distributions was checked with the Shapiro–Wilk test. Comparisons of variables between groups were performed using the Mann–Whitney U-test and the Kruskal–Wallis test, as appropriate. Statistical significance was established at *p* < 0.05. Data were analyzed using SPSS 23.0 for Windows (SPSS Inc., Chicago, IL, USA).

## 3. Results

We enrolled 132 atopic patients (92 women, 69.7%) from Policlinico Universitario “D. Casula” in Monserrato (Cagliari) and Ca’ Foncello Hospital in Treviso. A total of 62 patients were receiving biological therapy during COVID-19 infection. The median age was 56 (IQR 22.8) for patients on biologicals and 48 (IQR 26.5) for those not on biologicals (*p* = 0.028). Patients not on biologicals were significantly younger than patients on biologicals; however, in the latter group of older patients, we did not observe a worse clinical course [17]. 

The two baseline groups were comparable in terms sex, body mass index (BMI), smoking history, and systemic oral corticosteroids use (OCS), as shown in Table 1. There were no significant differences between the two groups in terms of non-biological therapy and comorbidity. The main comorbidities associated with the primary disease of interest were cardiovascular disease, such as hypertension, and autoimmune disease, such as type 2 diabetes mellitus and thyroid pathologies. The patients not on biological therapy had a prevalence of asthma of 87%, CRSwNP of 52%, CSU of 10%, and AD of 6%. The patients on biologicals had a prevalence of asthma of 93%, CRSwNP of 17%, CSU of 10%, and AD of 4%, as shown in Table 2. Disease severity matching was not possible between the two groups, because it is the main criterion for biological drug prescription. In our study, 62 participants were receiving biological therapy at the time of infection, as described in Table 2: anti-IgE (omalizumab) 30.6%, anti-IL5 (mepolizumab) 25.8%, anti-IL5R (benralizumab) 19.4%, and anti-IL4/IL13R (dupilumab) 24.2%. Nineteen patients on biological therapy (19/62, 30.6%) received omalizumab during COVID-19 infection. Thirteen patients received omalizumab for severe asthma at a dose of 150–650 mg, depending on the weight and basal IgE level; five of them had CRSwNP and one of them also had AD. Six patients received omalizumab for CSU at a dose of 300 mg every month; one of them had mild asthma as a comorbidity. Sixteen patients on biological therapy (16/62, 25.8%) were receiving mepolizumab during COVID-19 infection for severe asthma at a dose of 100 mg every four weeks; nine of them had CRSwNP as a comorbidity. Twelve patients on biological therapy (12/62, 9.4%) were receiving benralizumab during COVID-19 infection for severe asthma at a dose of 30 mg every eight weeks after the induction period in the first three months of treatment; six of them had CRSwNP as a comorbidity, and one of them had AD. Fifteen patients on biological therapy (15/62, 24.2%) were receiving dupilumab during COVID-19 infection. Eight patients received dupilumab with an indication for severe asthma at a dose of 300 mg every two weeks with induction of 300 mg plus 300 mg in the first administration; four of them had CRSwNP as a comorbidity, and one of them had AD. Six patients were receiving dupilumab with an indication for CRSwNP at a dose of 300 mg every two weeks; four of them had mild asthma as a comorbidity. One patient received dupilumab with an indication for AD at a dose of 300 mg every two weeks with induction of 300 mg plus 300 mg in the first administration.

### 3.1. Clinical Characteristics of SARS-CoV-2 Infection

We evaluated the duration of swab positivity, measured by rt-PCR methods, and the duration of symptoms in days. We did not find a significant difference. The median duration of swab positivity was 11 days (IQR 7) for patients on biologicals and of 12 days (IQR 4.8) in the control group (*p* = 0.45), as shown in Figure 1. In terms of duration of symptoms, there was not a significant difference between the two groups (*p* = 0.38); the medians were 5 days (IQR 7) and 5 days (IQR 7), respectively, in patients on biologicals and not on biologicals. We then collected information about the severity of symptoms; fifteen patients experienced an asymptomatic infection. We analyzed the relative frequency of fever (*p* = 0.63), asthenia (*p* = 0.19), myalgia (*p* = 0.16), anosmia and ageusia (*p* = 0.31), diarrhea (*p* = 0.19), sore throat (*p* = 0.06), cough (*p* = 0.66), pneumonia (*p* = 0.39), dyspnea (*p* = 0.016), and common cold symptoms (*p* = 0.038) between patients on biologicals and patients not on biologicals. During COVID-19 infection, patients on biologicals experienced significantly more frequent common cold-like symptoms and dyspnea. 

### 3.2. Asthma Exacerbation

The majority of patients (90.2%) had a previous diagnosis of asthma. We compared the asthma exacerbations between groups. Patients on biologicals experienced more exacerbations with respect to the control group; however, this increase was not significant (*p* = 0.21), as shown in Figure 2. Then we analyzed the exacerbation rates comparing the different biological drug classes, but we did not find significant differences (*p* = 0.97). 

### 3.3. Hospitalization

During the data collection period, five patients were hospitalized for a moderate or severe COVID-19 infection, according to the World Health Organization (WHO) classification, requiring oxygen supplementation. The rate of hospitalization was 4/70 for patients not on biologicals and 1/62 for patients on biologicals. All hospitalized patients had not yet received the COVID-19 vaccination at the time of hospitalization; they all had a good outcome and were discharged to their home in the absence of further complications and made a complete recovery. 

### 3.4. Vaccination Status

We stratified the patients according to vaccination status. Among patients that had received at least one dose of the vaccine (*n* = 96), the median duration of swab positivity of biological patients (*n* = 47) was 10 days (IQR 6.8), and it was 10 days (IQR 7) in the control group (*n* = 49); the median duration of symptoms in biological patients was 5 days (IQR 6.5), and it was 5 days (IQR 6.3) in the control group. Between the two groups, in those patients that received at least one dose of the vaccine, there was no significant difference in the duration of swab positivity (*p* = 0.94) and duration of symptoms in days (*p* = 0.77), as shown in Table 3. Among unvaccinated patients (*n* = 36), the median duration of swab positivity in unvaccinated patients on biologicals (*n* = 15) was 15 days (IQR 5.8), and it was 19 days (IQR 12.3) in the control group (*n* = 21); the median duration of symptoms in unvaccinated biological patients was 10 days (IQR 5.8), and it was 10 days (IQR 11) in the control group. Between all unvaccinated patients, there were no significant differences in the duration of swab positivity (*p* = 0.24) (shown in Figure 3) and duration of symptoms in days (*p* = 0.65). We also observed an increased number of hospitalizations in biological unvaccinated patients (*n* = 1/15) and in patients not on biologicals (*n* = 4/21) compared to patients that received at least one dose of the vaccine, among which there were no hospitalizations in both vaccinated patients on biologicals (*n* = 0/47) and in vaccinated patients not on biologicals (*n* = 0/49). The difference in hospitalization rate was not significant (*p* = 0.28); it was 4/21 (19%) for unvaccinated patients not on biological therapy and 1/15 (0.67%) for unvaccinated patients on biologicals. 

### 3.5. Safety Data

None of the patients involved in the study reported a potential relevant safety issue or adverse effect associated with the use of biological treatments during or after COVID-19 infection.

## 4. Discussion

In this study, we found that in our cohort of patients treated with mAbs targeting Th2 inflammation, which usually continued during COVID-19 infection, there were no signals of prolongation of infection or more severe respiratory symptoms or worse outcomes.

### 4.1. The Possible Link between Type 2 Inflammation, COVID-19, and mAbs

The severity of manifestations in patients with COVID-19 mostly depends on host immune response to infection [2]. Severe COVID-19 seems to correlate with an early defective Th1 response linked to an increased Th2 response in the context of a cytokine dysregulation, with an increase in interleukin (IL)-13, IL-5, eotaxin-2, IgE, and eosinophils [18,19]. IgE may play an immunomodulatory role in the immune response, activating basophils and mast cells, with a subsequent degranulation with cytokine release (e.g., IL-1, IL-6, TNF-α, IFN), complement/coagulation activation, and regulation of T cell responses [20]. Eosinophils may also play a role in inflammation, as they are resident in tissues and migrate during respiratory infections, exerting antiviral functions. However, in the more advanced stages of COVID-19, they can complicate the immune response to the virus [21]. It is tempting to speculate that biological therapies that can modulate the activity of eosinophils, IgE, and type 2 cytokines may dampen a part of the inflammatory response to SARS-CoV-2 infection. Omalizumab, for example, binds free serum IgE and prevents its binding to IgE receptors on mast cells, reducing degranulation and mediator release [22]. Mepolizumab and benralizumab, through IL-5 signal blocking, promote a reduction in eosinophils. Dupilumab may modify the response to viruses exerting an inhibitory action on the IL-13 pathway, which, when increased, is associated with a downregulation of the ACE2 receptor, thus reducing SARS-CoV-2 penetration into the lung and nasopharynx [5]. Through the action of IL-13, Th2 inflammation was also shown to dramatically upregulate TMPRSS2 in 695 children’s nasal airway transcriptome data; nevertheless, the number of cells that simultaneously express TMPRSS2 and ACE2 is quite small. Consequently, it is debated if patients with allergic asthma who express higher levels of TMPRSS2 are more susceptible to SARS-CoV-2 infection [4]. In another study, it was hypothesized that omalizumab, mepolizumab/benralizumab, and dupilumab may boost Th1 responses, and this may lead to an enhanced viral clearance by the adaptive immune system [23,24].

The possibility of continuing mAbs during infections such as COVID-19 may also reduce the risk of flares among these patients with severe disease [25,26,27], indirectly limiting the use of courses of systemic glucocorticoids that are a risk factor for infectious complications [28,29,30,31].

### 4.2. Asthma Patients

The results of our work are consistent with the outcomes of other studies previously published, extending the observation period through the Omicron wave. Heffler et al., at the beginning of the pandemic, evaluated the progress of COVID-19 in patients with severe asthma who were part of the Severe Asthma Network in Italy (SANI). They showed that among the 978 patients with severe asthma undergoing biological therapy, only 21 patients had COVID-19, and even if a fatal outcome was observed, the mortality rate for COVID-19 in this cohort of patients turned out to be lower than that of the estimates for that period in the general population (7.7% versus 14.5%). They suggested that biological treatment did not increase the risk of mortality in the course of COVID-19 [32]. In our asthmatic patients, we found no difference in the incidence of severe COVID-19 between asthmatic patients treated with biologicals and patients not receiving biologicals (*p* > 0.05), even when we stratified for different classes of biologicals. In terms of the duration of symptoms and swab positivity, there was not a significant difference between the two groups. Patients on biologicals experienced more asthma exacerbations with respect to the control group, but this increase was not significant and is partially influenced by the fact that these patients have more severe forms of asthma (the main criterion for starting biologicals) often associated with multiple comorbidities (such as uncontrolled MRGE, diabetes mellitus, and cardiovascular disease) that may act as independent risk factors for exacerbations. This is in line with data coming from the Dutch Severe Asthma Registry RAPSOD, showing that asthmatic patients hospitalized for COVID-19 had a high number of recognized risk factors to increase the risk of severe forms: obesity, diabetes mellitus, and cardiovascular disease, similar to the patients we reported here [33]. During our data collection period, five patients were hospitalized and all had good outcomes; only one of them was on biologicals (omalizumab). Interestingly, a recent study by Michelle Le et al. investigated the effectiveness and safety of omalizumab in adults hospitalized for COVID-19 pneumonia. This randomized, double-blind, controlled phase II trial showed that omalizumab may be effective in reducing progression to mechanical ventilation or death at day 14 and also useful for reducing all-cause mortality on day 28. This study also provides the first clinical evidence for a possible antiviral and/or anti-inflammatory effect of omalizumab in non-atopic patients hospitalized for a severe viral respiratory disease [34].

No patients died of COVID-19 during our data collection period. Our findings are in line with data collected in 2020 in a cohort enrolled by The Belgian Severe Asthma Registry (BSAR). This study showed that treatment with mAbs for severe asthma was not associated with an increased risk of infection by SARS-CoV-2 or more severe COVID-19. Also, there was no difference in COVID-19 incidence between severe asthma patients treated with biologicals (anti-IgE or anti-IL5/anti-IL5R) and patients not receiving any asthma biological (*p* > 0.05) [35]. In 2021, Poddighe et al. attempted to summarize the available literature on the use of anti-type 2 inflammation mAbs during SARS-CoV-2 infection, their impact on the course of the disease, and the likelihood of developing COVID-19, concluding that these treatments can be used safely, neither improving clinical outcomes nor increasing susceptibility, and may even have a protective effect on severe forms [36].

### 4.3. Patients with Chronic Rhinosinusitis with Nasal Polyposis (CRSwNP)

In the literature, the first report of COVID-19 in a patient with CRSwNP treated with dupilumab showed a mild clinical course [37]. Later, some case reports and small series and also expert opinion confirmed that the use of dupilumab in patients with CRSwNP was safe and should be continued during SARS-CoV2 infection [38]. Patients of our cohort with CRSwNP have paralleled the trend of asthma patients and no patient with CRSwNP was hospitalized. During the study, all the patients on biologicals continued the therapy with dupilumab without reports of worsening in SNOT-22 or developing a severe COVID-19 outcome.

### 4.4. Atopic Dermatitis (AD) Patients

For AD, since the beginning of the pandemic, the data about biologicals were reassuring, in particular for the use of dupilumab. Ferrucci et al. reported two adult patients treated with dupilumab for severe AD who did not stop biological therapy at the time of diagnosis of COVID-19, and despite this, they had a mild clinical course [39]. Chiricozzi et al. published a case series of COVID-19 patients treated with dupilumab in Italy [40]. In a total of 1831 Italian adult patients with moderate or severe AD, 16 cases of COVID-19 were identified, and of those, 15 were receiving dupilumab. During COVID-19 infection, eight subjects continued to use dupilumab, whereas the others stopped the drug. COVID-19-related complications were not reported in both groups. Based on these findings, patients enrolled in our study who received dupilumab for AD continued the therapy during infection without experiencing any exacerbation or a severe COVID-19-related outcome. 

### 4.5. Chronic Spontaneous Urticaria (CSU) Patients

The first data on the safety of ongoing biological therapy for COVID-19 in patients with CSU date back to the work of Passante et al., in which they studied seven adult patients with CSU undergoing biological treatment during SARS-CoV-2 infection; none of them had to interrupt the biological therapy and three patients were even asymptomatic [41]. Patients enrolled in our study for CSU being treated with omalizumab did not experience exacerbations or other complications compared to the control group of patients who had CSU not treated with biologicals, and none needed hospitalization.

### 4.6. Vaccination Status

Focusing on the vaccination status of our cohort, the data show that in those patients who were not vaccinated at the time of infection, the duration of symptoms and the positivity of the swab were longer, even though not statistically significant, with an increase in mild-to-moderate forms of COVID-19. 

## 5. Conclusions

Our investigation showed that anti-type 2 inflammation mAbs during COVID-19 infection are safe, and did not worsen the clinical course or outcomes. No patients on biologicals reported safety issues or adverse effects associated with the use of this drug class during COVID-19 infection. 

All these findings need to be evaluated, considering the limitations of the study. The most noteworthy limitation is represented by the observational real-life design of this retrospective study. Also, the sample size, the limited ethnic diversity, and the absence of a longer follow-up period must be taken into consideration. 

The results of this study suggest that biological therapy against type 2 inflammation is safe, in keeping with the available scientific literature. Therefore, therapy should routinely not be discontinued during infection, and this should be associated with the best practices for COVID-19 management, including vaccination. Controlled studies and registries of these biologicals and more extensive real-life studies are required to draw firm conclusions about safety and potential benefits that anti-type 2 inflammation mAbs could offer in particular clinical situations, such as infections.

## Figures and Tables

**Figure 1 life-14-00378-f001:**
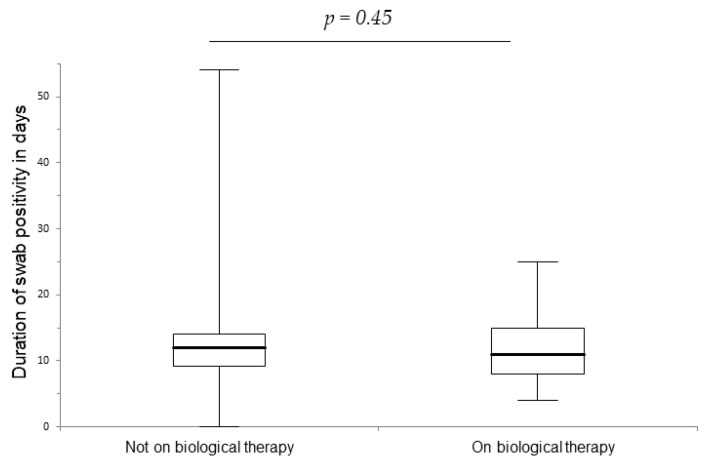
Duration of swab positivity in days.

**Figure 2 life-14-00378-f002:**
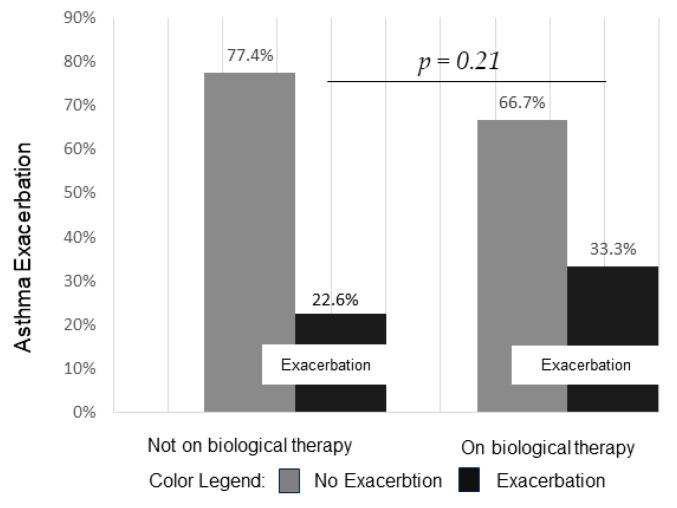
Relative frequency of asthma exacerbation.

**Figure 3 life-14-00378-f003:**
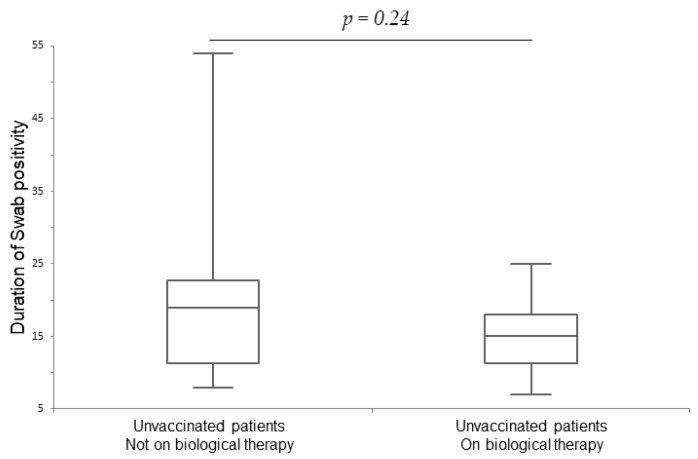
Unvaccinated patients: duration of swab positivity in days.

**Table 1 life-14-00378-t001:** Baseline characteristics of patients.

Characteristic	On Biological *n* = 62 (47%)	Not on Biological *n* = 70 (53%)	*p*-Value
Age (years) Median ± IQR *	56 ± 22	48 ± 26	0.028
Female (%)	47	53	0.93
Never smoker (%)	40	44	0.15
BMI Median ± IQR ^†^	24 ± 5.93	24.10 ± 4.49	0.66
OCS ^‡^ mg/day Median ± IQR *	7.5 ± 6.46	5 ± 16.25	0.92
Current on OCS ^‡^ (%)	17.7	18.5	/

Data are presented as number (%) or median (* interquartile range (IQR)). ^†^ BMI: body mass index; ^‡^ OCS: systemic oral corticosteroids.

**Table 2 life-14-00378-t002:** Prevalence of disease and frequency of biological therapy.

Cohort of Patients	On Biological *n* = 62 (47%)	Not on Biological *n* = 70 (53%)
Asthma (%)	87	93
Chronic Rinosinusites with Nasal Polyposis (%)	52	17
Chronic Spontaneous Urticaria (%)	10	10
Atopic Dermatitis (%)	6	4
Omalizumab (%)	30.6	/
Mepolizumab (%)	25.8	/
Benralizumab (%)	19.4	/
Dupilumab (%)	24.2	/

Data are presented as number (%).

**Table 3 life-14-00378-t003:** Vaccination Status.

	Vaccinated Patients on Biologicals (*n* = 47)	Vaccinated Patients Not on Biologicals (*n* = 49)	*p*-Value	Unvaccinated Patients on Biologicals (*n* = 15)	Unvaccinated Patients Not on Biologicals (*n* = 21)	*p*-Value
Duration of swab positivity in days	10 ± 6.8 *	10 ± 7 *	0.94	15 ± 5.8 *	19 ± 12.3 *	0.24
Duration of symptoms in days	5 ± 6.5 *	5 ± 6.3 *	0.77	10 ± 5.8 *	10 ± 11 *	0.65

Data are presented as median (* interquartile range (IQR)).

## Data Availability

Data will be made available upon reasonable request from the corresponding author.

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
