# Peer review of "COVID-19 Clinical Features and Outcome in Italian Patients Treated with Biological Drugs Targeting Type 2 Inflammation"

_life, 2024, doi:10.3390/life14030378_

Round 1

Reviewer 1 Report

Comments and Suggestions for Authors

A paper by Sambugaro et al. describes COVID-19 aspects in Italian patients treated with biological drugs targeting type 2 inflammation. The article is well-written and presents pretty interesting data. I have minor comments.

Please present a detailed background in the abstract on the topic you are discussing and delete words such as background, methods, results, and conclusion. More practical information should be included at the end of the abstract.

SARS-CoV-2, Th2, please change into this form. Lines 96-107, references?

The questionnaire should be published in the supplementary materials.

Why did you assess the normality with the KS test, not Shapiro-Wilk one? How did you present non-normally distributed data? Should be median with Q1-Q3, please change it throughout the text.

Line 131, 92 women, ?%.

Generally, statistically significant p-values should be presented rounded to the third decimal place, or p<0.001. Not significantly results to the second decimal place. Please mark statistically significant results.

Tables. Categorical data should be presented as numbers with %.

Figures can be omitted, there are no differences between analyzed groups. This information can be shown in the text.

Line 178. Which groups were compared?

The discussion is disorganized. Please divide it into smaller paragraphs to increase the clarity.

Conclusions. More practical information is demanded.

Line 334, please delete. Line 335?

Comments on the Quality of English Language

Minor editing of English language required.

Reviewer 2 Report

Comments and Suggestions for Authors

The article entitled "COVID-19 Clinical Features and Outcome in Italian Patients Treated with Biological Drugs Targeting Type 2 Inflammation" (life-2890567) is being presented in the "Epidemiology" section of the Special Issue "COVID-19 Prevention and Treatment: 2nd Edition." The submitted article is well-suited for this section.

This is a multicentric investigation involving two centers that examined the clinical course of COVID-19 in patients receiving biological therapy targeting type 2 inflammation and those not receiving biologicals.

Comments:

  1. In the abstract, the study's design should be specified. The objective statement should align more closely with the methodology of the results, especially considering the inclusion of vaccination status, which is not reflected in the objective. The study design is reflected in the conclusion, whereas the conclusion should be the most significant contribution of the results. Given that the abstract is a crucial element accessible on various platforms, it should be as specific and comprehensive as possible for potential readers to understand the objective and achievements.
  2. The introduction is clear and concise, emphasizing the importance of the topic. However, it is suggested that it could benefit from a more thorough review of the literature on this topic. Additionally, the objective, as a general summary, should be more in line with the methodology and results presented.
  3. Regarding the methodology, it is clearly presented. However, it is not mentioned whether a sample size calculation has been performed. If conducted, it should be included. If not, and a convenience sampling approach was used, the study's power should also be specified to assess the results accurately. The participation rate of patients meeting the inclusion criteria should be indicated.
  4. In the discussion, the first paragraph typically presents the most relevant results of the study. However, the authors discuss the study's design, which should be included in the materials and methods section. Biochemical parameters mentioned in the discussion should be presented in the results section.

The limitations of the sample should be considered concerning the conclusions and, consequently, prompt a thoughtful discussion.

Reviewer 3 Report

Comments and Suggestions for Authors

In order to improve the quality of the document and to address concerns, the following expanded points and additions are suggested:

1. Diversity of patient population: The study could benefit from the inclusion of a more diverse patient population with different demographic characteristics and geographic areas. This diversity would help to understand the efficacy and safety of biological treatments in a broader spectrum of the overall population, making the results more generalisable.

2. Longitudinal data collection: Extending the duration of follow-up would provide valuable insights into the long-term effects and sustainability of biological treatments on COVID-19 outcomes. This could highlight potential late emergencies. This could highlight potential late effects or long-term benefits that are not evident in shorter duration studies.

3. Control group design: Improving the control group design to ensure a close match with the treatment group in terms of demographics, disease severity and other relevant characteristics could strengthen the validity of the study. This would help to make more accurate comparisons between the treatment and control group and reduce potential bias.

4. Mechanistic insights: The article could be improved by providing mechanistic insights into how biological treatments interact with the pathogenesis of COVID-19. A more in-depth discussion of the underlying biological mechanisms could enrich the understanding of why certain treatments are effective or ineffective, providing a more complete view of the therapeutic landscape.

5. Reporting of safety data: There is a need for more detailed information on the safety and adverse effects associated with the use of biological treatments in patients with COVID-19. Greater transparency in this area would inform patients about the safety of treatments. Greater transparency in this area would inform physicians and patients about potential risks, contributing to more informed decision-making in clinical practice.

6. Depth of discussion: The discussion section appears superficial and shallow in exploring the implications, limitations and potential for future research based on the study results. A more thorough analysis and contextualisation of the findings within the existing body of knowledge would greatly enhance the contribution of the work to the field.

7. Plagiarism issues: Address plagiarism issues as outlined in the report provided. It is crucial to ensure that all content is original or properly cited, maintaining the integrity and credibility of the research. Rewriting plagiarised sections and inserting appropriate citations where necessary are essential steps to correct this problem.

Round 2

Reviewer 2 Report

Comments and Suggestions for Authors

I have carefully reviewed the new version of the manuscript titled "COVID-19 Clinical Features and Outcome in Italian Patients Treated with Biological Drugs Targeting Type 2 Inflammation" (life-2890567), as well as the authors' response to the comments provided. I believe that the authors have enhanced the quality of their work. However, there remains a significant issue with the sample size, a concern that the authors now acknowledge in the discussion and conclusions sections.

Reviewer 3 Report

Comments and Suggestions for Authors

The manuscript has improved a lot. The authors addressed all the shortcomings.